

# A visual analytic approach for the identification of ICU patient subpopulations using ICD diagnostic codes

Daniel Alcaide[1] and Jan Aerts[1,2]

[1] Department of Electrical Engineering (ESAT) STADIUS Center for Dynamical Systems, Signal Processing and Data Analytics, KU Leuven, Leuven, Belgium
[2] UHasselt, I-BioStat, Data Science Institute, Hasselt, Belgium

## ABSTRACT

A large number of clinical concepts are categorized under standardized formats that ease the manipulation, understanding, analysis, and exchange of information. One of the most extended codifications is the International Classification of Diseases (ICD) used for characterizing diagnoses and clinical procedures. With formatted ICD concepts, a patient profile can be described through a set of standardized and sorted attributes according to the relevance or chronology of events. This structured data is fundamental to quantify the similarity between patients and detect relevant clinical characteristics. Data visualization tools allow the representation and comprehension of data patterns, usually of a high dimensional nature, where only a partial picture can be projected. In this paper, we provide a visual analytics approach for the identification of homogeneous patient cohorts by combining custom distance metrics with a flexible dimensionality reduction technique. First we define a new metric to measure the similarity between diagnosis profiles through the concordance and relevance of events. Second we describe a variation of the Simplified Topological Abstraction of Data (STAD) dimensionality reduction technique to enhance the projection of signals preserving the global structure of data. The MIMIC-III clinical database is used for implementing the analysis into an interactive dashboard, providing a highly expressive environment for the exploration and comparison of patients groups with at least one identical diagnostic ICD code. The combination of the distance metric and STAD not only allows the identification of patterns but also provides a new layer of information to establish additional relationships between patient cohorts. The method and tool presented here add a valuable new approach for exploring heterogeneous patient populations. In addition, the distance metric described can be applied in other domains that employ ordered lists of categorical data.

Corresponding author
Jan Aerts, jan.aerts@uhasselt.be

## INTRODUCTION

Patient profiling and selection are a crucial step in the setup of clinical trials. The process involves analytical methods to handle the increasing amount of healthcare data but is still

extremely labor-intensive (*Sahoo et al., 2014*). Nevertheless, the input from an expert in this selection is important.

To support the expert in the selection of suitable patients, visual analytics solutions can enable the exploration of a patient population, make recruitment consistent across studies, enhance selection accuracy, increase the number of selected participants, and significantly reduce the overall cost of the selection process (*Fink et al., 2003*; *Damen et al., 2013*). Visual analytics relies on interactive and integrated visualizations for exploratory data analysis in order to identify unexpected trends, outliers, or patterns. It can indicate relevant hypotheses that can be complemented with additional algorithms, and help define parameter spaces for these algorithms (*Franken, 2009*). A major challenge in creating visual solutions is to find effective tools which allow the projection of all data dimensions. One popular solution is to visualize the relationship between elements rather than raw data through similarity metrics which quantify the closeness between data objects (*Liu et al., 2016*). Similarity metrics are a fundamental part for most of the case-based reasoning algorithms (*Kolodner, 2014*) such as the detection of consistent cohorts of patients within a patient population. One of the remaining open challenges in the analysis of patient similarity is to establish relevant and practical ways based on clinical concepts (*Jia et al., 2019*).

Many types of information about the patient profile such as diagnosis, procedures, and prescriptions are available under standardized categories contained in taxonomies or dictionaries, e.g., the International Classification of Diseases (ICD), Medical Dictionary for Regulatory Activities (MedDRA) and the Anatomical Therapeutic Chemical (ATC) Classification System. Each patient is for example linked to an ordered list of diagnoses, which are semantic concepts that are (in the case of MIMIC (*Johnson et al., 2016*)) ordered from most to least important (as per the MIMIC-III documentation "ICD diagnoses are ordered by priority—and the order does have an impact on the reimbursement for treatment"). These standardized formats provide a non-numerical data structure facilitating both understanding and management of the data. Several methods have been proposed to define similarity between lists of clinical concepts based on presence of absence of specific terms (*Gottlieb et al., 2013*; *Zhang et al., 2014*; *Brown, 2016*; *Girardi et al., 2016*; *Rivault, Le Meur & Dameron, 2017*; *Jia et al., 2019*). However, the diagnostic profile of a patient is not merely an independent list of semantic concepts but also includes an intrinsic order indicated by the position of the terms in the list reflecting the relevance vis-a-vis the actual patient status. To the best of our knowledge, no previous work has combined the categorical and ordinal nature of clinical events into a single distance function. This dualism can contribute to improving the detection of cohorts through diagnostic and procedural data. This can significantly impact clinical trials when diagnoses or procedures are part of the recruitment criteria (*Boland et al., 2012*).

In this paper, a novel approach for exploring clinical patient data is introduced. In particular, we focus on patient profiles represented by a set of diagnosis ICD codes sorted by relevance. The distance metric considers the sorted concepts as input, and the resulting pairwise values are used to create a graph where similar patients are connected.

The remaining part of this paper is organized as follows. In the section "Background", we give an overview of related work in categorical events and graphical projections of patient similarity. The section "Materials and Methods" describes the proposed distance metric and modifications applied on the base algorithms STAD for visualizing patient population. In "Results", we demonstrate the effectiveness of the approach in a real-world dataset. The section "Discussion" compares other methods and alternative metrics for similar data. Finally, the section "Conclusion" presents conclusions and possible directions for future work.

## BACKGROUND

The exploration and analysis of patients through similarity measures has been presented in different areas of bioinformatics and biomedicine, and also data mining and information visualization. In this section, we review the related literature on these areas below, and we focus on the notion of similarity measures for categorical events and graphical representation of patient similarity.

### Patient similarity and distance measures for categorical events

Different distance metrics exist for unordered lists of categorical data, including the overlap coefficient (*Vijaymeena & Kavitha, 2016*), the Jaccard index (*Real & Vargas, 1996*), and the simple matching coefficient (*Šulc & Řezanková, 2014*). These methods compute the number of matched attributes between two lists using different criteria. Although they treat each entry in the list as independent of the others, they have been used successfully to measure patient similarity to support clinical decision making and have demonstrated their effectiveness in exploratory and predictive analytics (*Zhang et al., 2014*; *Lee, Maslove & Dubin, 2015*). Similarly, different ways of computing distances between ordered lists are available (*Van Dongen & Enright, 2012*). The Spearman's rank coefficient (*Corder & Foreman, 2014*) is useful for both numerical and categorical data and has been used in clinical studies (*Mukaka, 2012*). However, correlation between ordered lists cannot be calculated when the lists are of different lengths (*Pereira, Waxman & Eyre-Walker, 2009*).

In the context of medical diagnoses, the International Classification of Diseases (ICD) codes have been widely used for describing patient similarity. However, these typically consider the hierarchical structure of the ICD codes. *Gottlieb et al. (2013)*, for example, proposed a method combining the Jaccard score of two lists with the nearest common ancestor in the ICD hierarchy. The similarity measure for the ICD ontology was previously presented in *Popescu & Khalilia (2011)*. Each term is assigned to a weight based on its importance within the hierarchy, which was defined as $1 - 1/n$ where $n$ corresponded to its level in the hierarchy.

In our work, however, we will not leverage the hierarchical structure of the ICD codes, but employ the ICD grouping as described by *Healthcare Cost & Utilization Project (2019)*. Our approach takes the position of the term in the list of diagnoses into account, which is a proxy to their relevance for the patient status. The metric assigns a higher weight to terms located earlier in the list.

Alternative approaches such as those by *Le & Ho (2005)* and *Ahmad & Dey (2007)* consider two elements similar if they appear together with a high number of common attributes. They must share the same relationships with other elements in the sample. The latent concept of these metrics is to find groups of co-occurrence such as the identification of disease comorbidities (*Moni, Xu & Lio, 2014*; *Ronzano, Gutiérrez-Sacristán & Furlong, 2019*) although these studies aim to find heterogeneous types of diseases rather than different profiles of patients. The main drawback of metrics based on co-occurrence is the assumption of an intrinsic dependency between attributes without considering their relevance. The work presented by *Ienco, Pensa & Meo (2012)* and *Jia, Cheung & Liu (2015)* use the notion of contexts to evaluate pairs of categories. A context is an additional dimension used to determine the similarity between pairs. If the context is another categorical dimension, the similarity between the two categories is determined by the resulting co-occurrence table's correlation.

## Graphical projections of patient similarity

Visually representing pairwise distance matrices remains a challenge. Most often, dimensionality reduction techniques are used to bring the number of dimensions down to two so that the data can be represented in a scatterplot (*Nguyen et al., 2014*; *Girardi et al., 2016*; *Urpa & Anders, 2019*). Such scatterplots can not only indicate clusters and outliers, but are also very useful for assessing sample quality. In the case of patient data, each point in such plot represents a patient, and relative positions between them in the 2D plane correspond to the distance between them in the original higher dimensional space. Multidimensional scaling (MDS) is arguably one of the most commonly used dimensionality reduction methods (*Mukherjee, Sinha & Chattopadhyay, 2018*). It arranges points on two or three dimensions by minimizing the discrepancy between the original distance space and the distance in the two-dimensional space. Since its first use, many variations of classical MDS methods have been presented, proposing modified versions of the minimization function but conserving the initial aim (*Saeed et al., 2018*). Besides MDS, recent methods have been proposed to highlight the local structure of the different patterns in high-dimensional data. For example, t-distributed stochastic neighbor embedding (t-SNE) (*Maaten & Hinton, 2008*) and uniform manifold approximation (UMAP) (*McInnes, Healy & Melville, 2018*) have been used in many publications on heterogeneous patient data (*Abdelmoula et al., 2016*; *Simoni et al., 2018*; *Becht et al., 2019*). Unlike MDS, t-SNE projects the conditional probability instead of the distances between points by centering a normalized Gaussian distribution for each point based on a predefined number of nearest neighbors. This approach generates robustness in the projection, which allows the preservation of local structure in the data. In a similar fashion, UMAP aims to detect the local clusters but at the same time generates a better intuition of the global structure of data.

In addition to scatterplot representations, alternative visual solutions are also possible, for example heatmaps (*Baker & Porollo, 2018*), treemaps (*Zillner et al., 2008*), and networks. The latter are often built using a combination of dimensionality reduction and topological methods (*Li et al., 2015*; *Nielson et al., 2015*; *Dagliati et al., 2019*).

This approach has for example been used with success to visually validate the automated patient classification in analytical pipelines (*Pai & Bader, 2018*; *Pai et al., 2019*). In general, the created network encodes the distance between two datapoints in high-dimensional space into an edge between them and the full dataset can therefore be represented as a fully connected graph. The STAD method (*Alcaide & Aerts, 2020*) reduces the number of edges allowing a more scalable visualization of distances. The original distance in high-dimensional space between two datapoints is correspondent to the path-length in the resulting graph between these datapoints. The main advantage of networks to display high-dimensional data is that users not only can perceive patterns by the location of points but also by the connection of elements, thereby increasing trust in the data signals.

## MATERIALS AND METHODS

The International Classification of Diseases (ICD) is a diagnosis and procedure coding system used by hospitals to bill for care provided. They are further used by health researchers in the study of electronic medical records (EMR) due to the ease of eliciting clinical information regarding patient status. Although these administrative databases were not designed for research purposes, their efficiency compared to the manual review of records and demonstrated reliability of information extracted have democratized the analysis of health data in this way (*Humphries et al., 2000*). Even though ICD codification is hierarchically organized, some concepts in the database may be under-reported (*Campbell et al., 2011*). To make analysis feasible, the ICD codes are in practice often grouped in higher categories to reduce noise and facilitate the comparison and analysis with automatic systems (*Choi et al., 2016*; *Miotto et al., 2016*; *Baumel et al., 2018*). In our approach, we adopt the ICD generalization introduced by the Clinical Classification Software (CSS) which groups diseases and procedures into clinically meaningful sections (*Healthcare Cost & Utilization Project, 2019*). Here we introduce a method to compare unequal sets of ordered lists of categories and explore the different cohorts of patients through visual representations of data. This approach employs a custom distance metric presented in section "Diagnosis similarity and distances" within the visual analytics method as presented in section "Spanning Trees as Abstraction of Data".

### Diagnosis similarity and distances

In the MIMIC dataset which was used for this work (*Johnson et al., 2016*), each patient's diagnosis is a list of ICD codes, as exemplified in Table 1. The average number of concepts per profile in the MIMIC III dataset is 13 with a standard deviation of five. Diagnoses are sorted by relevance for the patient status. This order determines the reimbursement for treatment, and, from an analysis perspective, can help us to distinguish similar medical profiles even with different initial causes. The similarity metric presented in this work takes this duality into account and provides support for comparing profiles with an unequal length of elements.

The similarity between two patients (diagnosis profiles) A and B is based on which diagnoses (i.e., ICD9 codes) are present in both, as well as the position of these elements in

**Table 1 Diagnosis profiles of two patients with sepsis in the MIMIC-III database.** Diagnosis profiles of two patients with sepsis in the MIMIC-III database (HADM_ID: 115057 and 117154). The list of diagnoses presented in this table are simplified for illustrative purposes. The patients share many diagnoses, although the order is different. The position of a concept corresponds to its importance in describing the patient status, i.e., the first position is the most important pathology and the last the least relevant. Concepts in bold highlight the matches between the two patients.

| Patient A (115057) | | | Patient B (117154) | | |
|---|---|---|---|---|---|
| | ICD section | Label (ICD9) | | ICD section | Label (ICD9) |
| 1 | 996-999. | Infection and inflammatory reaction due to other vascular device, implant, and graft (99662) | 1 | 430-438. | Unspecified intracranial hemorrhage (4329) |
| 2 | **990-995.** | **Sepsis (99591)** | 2 | 430-438. | Cerebral artery occlusion, unspecified with cerebral infarction (43491) |
| 3 | **590-599.** | **Urinary tract infection, site not specified (5990)** | 3 | 996-999. | Iatrogenic cerebrovascular infarction or hemorrhage (99702) |
| 4 | **401-405.** | **Unspecified essential hypertension (4019)** | 4 | **990-995.** | **Sepsis (99591)** |
| | | | 5 | **590-599.** | **Urinary tract infection, site not specified (5990)** |
| | | | 6 | **401-405.** | **Unspecified essential hypertension (4019)** |

the list. Consider a match M between two concepts $c_A$ and $c_B$, which contributes to the similarity according to the following formula:

$$M_C(A, B) = \ln\left(1 + \frac{1}{\max(\text{position}(c_A), \text{ position}(c_B))}\right)$$

The position mentioned in the formula corresponds to the positional index in the list. As an example, the individual contribution of the concept "Sepsis" for patients A and B in Table 1 is $M_{\text{Sepsis}} = \ln\left(1 + \frac{1}{\max(2,4))}\right) = \ln 1.25$. The total similarity between patients is the sum of individual contributions from the matched concepts $S(X, Y) = \sum_n^{i=1} M(X \cap Y)$. Applying this formula to the example in Table 1 gives: $S$ (Patient A, Patient B) = $M_{Sepsis} + M_{\text{Urinary tract infection}} + M_{\text{Hypertension}} = \ln 1.25 + \ln 1.20 + \ln 1.17 \simeq 0.56$.

To perform the patient analysis in STAD (Section "Simplified Topological Abstraction of Data"), the similarity measure $S$ needs to be converted into a distance measure $D = 1 - S_{\text{normalized}}$ where $S_{\text{normalized}} = S/\max(S)$.

Distance measures in categorical variables are built based on a binary statement of zero or one. Unlike other data types, categorical data generate a bimodal distribution, which can be considered as a normal when the element contains multiple dimensions (*Schork & Zapala, 2012*). The similarity in diagnosis metric not only depends on the matching of elements but also on their positions on the list. These two conditions tend to generate left-skewed distance distributions, as shown in Fig. 1A. In other words, most patients are very different from other patients.

## Simplified topological abstraction of data

Simplified Topological Abstraction of Data (STAD) (*Alcaide & Aerts, 2020*) is a dimensionality reduction method which projects the structure of a distance matrix $D_X$ into a graph U. This method converts datapoints in multi-dimensional space into an unweighted graph in which nearby points in input space are mapped to neighboring vertices in graph space. This is achieved by maximizing the Pearson correlation between

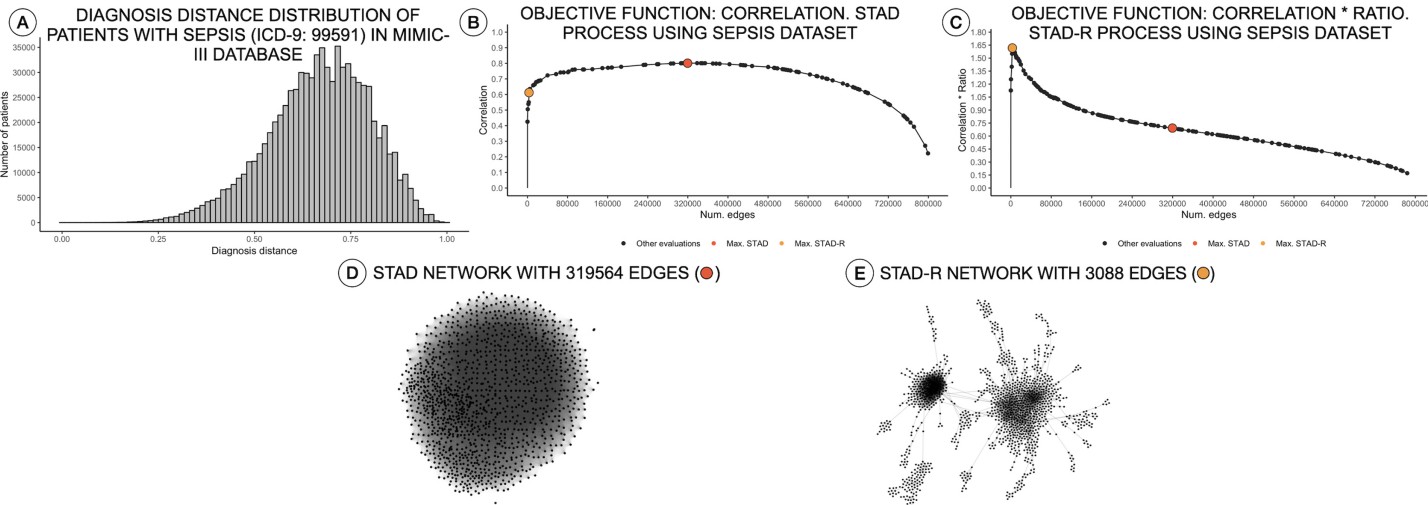

**Figure 1 Distance distributions of a population of patients with sepsis, STAD, and STAD-R projections.** The dataset is composed of a selection of 1,271 patients from MIMIC-III diagnosed with sepsis (ICD-9: 99591). Predefined conditions cause more homogeneous populations that mitigate the skewness of the diagnosis similarity distribution. (A) Distribution of diagnosis distance. (B) Correlation between original distance matrix and distance matrix based on STAD graph, given different numbers of edges. (C) Idem as (B) using STAD-R. (D) STAD network. (E) STAD-R network.

the original distance matrix and a distance matrix based on the shortest paths between any two nodes in the graph (which is the objective function to be optimized). STAD projections of multi-dimensional data allow the extraction of complex patterns. The input for a STAD transformation consists of a distance matrix of the original data, which in this case is based on the metric as defined in the previous section.

As mentioned above, high dissimilarity between datapoints (i.e., patients) results in a left-skewed distance distribution. Unfortunately, this skew poses a problem for STAD analysis. As mentioned above, the STAD method visualizes the distances between elements by means of the path length between nodes. Hence, to represent a big distance between two elements, STAD needs to use a set of intermediate connections that help to describe a long path. In case no intermediate nodes can be found, the algorithm forces a direct connection between the two nodes. As a result, in a left-skewed distribution, STAD tends to generate networks with an excessively high number of links, even when high correlation can be achieved as shown in Figs. 1B and 1D. This means that the principle that nodes that are closely linked are also close in the original space (i.e., are similar) does not hold anymore (*Koffka, 2013*).

Therefore, we propose a modification of the STAD algorithm, named STAD-R (where the R stands for "Ratio"), which avoids the problem on datasets of dissimilar items through the use of a modified objective function. To reduce the number of links between dissimilar datapoints we alter the STAD method to incorporate the ratio $R = \frac{\sum 1 - d_{\text{network edge}}}{\sum 1 + d_{\text{network edge}}}$, in which the sum of $d_{\text{network edge}}$ refers to the sum of distances of edges included in the network (see Fig. 2). Note that edges represent the distance between two elements of the dataset and constitute a cell in the pairwise distance matrix.
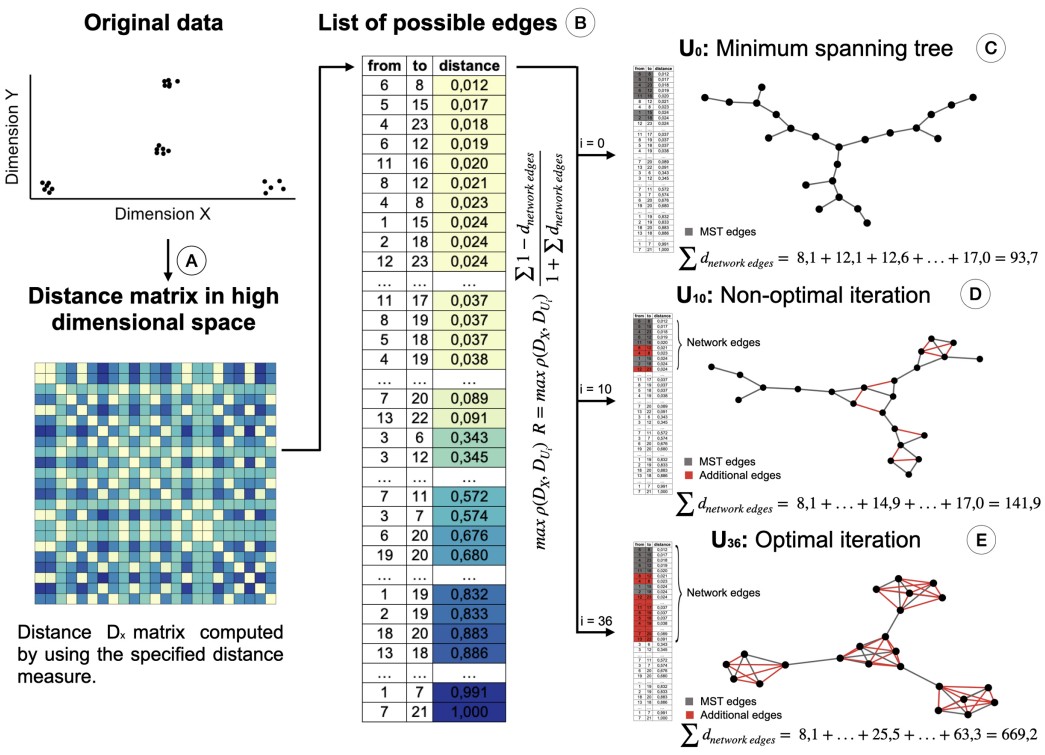

**Figure 2** **Creation of the STAD-R network for different iterations.** (A) Distance matrix DX : Pairwise distances between all elements in a point cloud are calculated using a defined distance metric. (B) Distance list: Transformation of the matrix into a edges list. Edges are sorted by their distance. Smaller distances are first candidates to become part of the network U. (C) The Minimum spanning tree connects all nodes with minimum distance. It guarantees that a path exists between all nodes and becomes the initial iteration in the evaluation of the optimal STAD network (D). The addition of edges over the MST may improve the correlation between the two distance matrices. Edges are added in sequential order following the list in B. (E) The optimal network is found at the iteration with the maximum combination of correlation between $D_X$ and $D_U$ and the ratio $R$.

This ratio $R$ is added to the objective function of the algorithm, which maximizes the correlation ρ between the distance matrices $D_X$ (of the input dataset) and $D_U$ (based on shortest path distances in the graph). When including the ratio $R$, the objective function in STAD-R is not only a maximization problem based on the Pearson correlation but also a maximization of ratio R. Table 1 shows the difference between STAD and STAD-R.

The ratio $R$ is the sum of those distances of datapoints in $D_X$ that are directly connected in network $U$. Figure 2 illustrates the creation of a STAD-R network during different iterations.

The result of STAD-R over STAD is presented in Fig. 1E. The network has considerably fewer links (Fig. 1C), and patterns in the data are much more apparent.

The STAD-R algorithm generates networks with considerably lower number of links compared to the correlation-based version. The ratio R restricts the inclusion of dissimilarities and therefore, the number of edges in the network. This new constraint also alters the number of edges in networks generated from other distributions types,

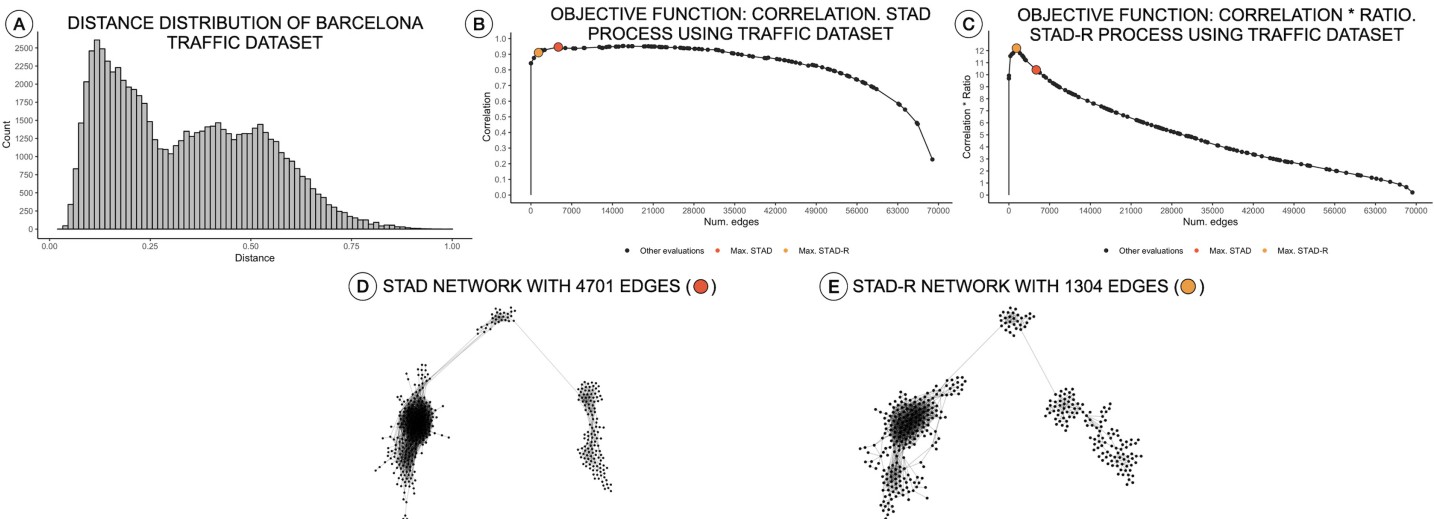

**Figure 3 Distance distributions of traffic activity, STAD, and STAD-R projections.** The dataset contains the traffic activity in the city of Barcelona from October 2017 until November 2018. The dataset was presented and analyzed in *Alcaide & Aerts (2020)* (A) Distribution of diagnosis distance. (B) Correlation between original distance matrix and distance matrix based on STAD graph, given different numbers of edges. (C) Idem as (B) using STAD-R. (D) STAD network. (E) STAD-R network.

e.g., right-skewed or normal. Nevertheless, the general "shape" of the resulting network remains the same. An example is presented in Fig. 3A, showing a right-skewed distance distribution, leading to networks with different numbers of edges for STAD and STAD-R, respectively. However, the structure is still preserved in both networks (Figs. 3D and 3E).

# RESULTS

We applied this approach to the MIMIC-III database (*Johnson et al., 2016*), which is a publicly available dataset developed by the MIT Lab for Computational Physiology, containing anonymized health data from intensive care unit admissions between 2008 and 2014. The MIMIC-III dataset includes the diagnosis profiles of 58,925 patients. Their diagnoses are described using the ICD-9 codification and sorted according to their relevance to the patient. To reduce the number of distinct terms in the list of diagnoses, ICD codes were first grouped as described in the ICD guidelines *Healthcare Cost & Utilization Project (2019)*. The proof-of-principle interface as well as the underlying code can be found on http://vda-lab.be/mimic.html.

The interface is composed of two main parts: an overview node-link network visualization including all patients (Fig. 4A), and a more detailed view of selected profile groups (Fig. 4B). Networks for each ICD code are precomputed: for each ICD-9 code the relevant patient subpopulations were extracted from the data, diagnosis distances and the resulting graph were computed using STAD-R. When the user selects an ICD-9 code from the interface (in this case code 2910; alcohol withdrawal delirium), the corresponding precomputed network is displayed.

The output of Louvain community detection (*De Meo et al., 2011*) is added as post-hoc annotation to facilitate the selection and exploration of the most evident patterns.

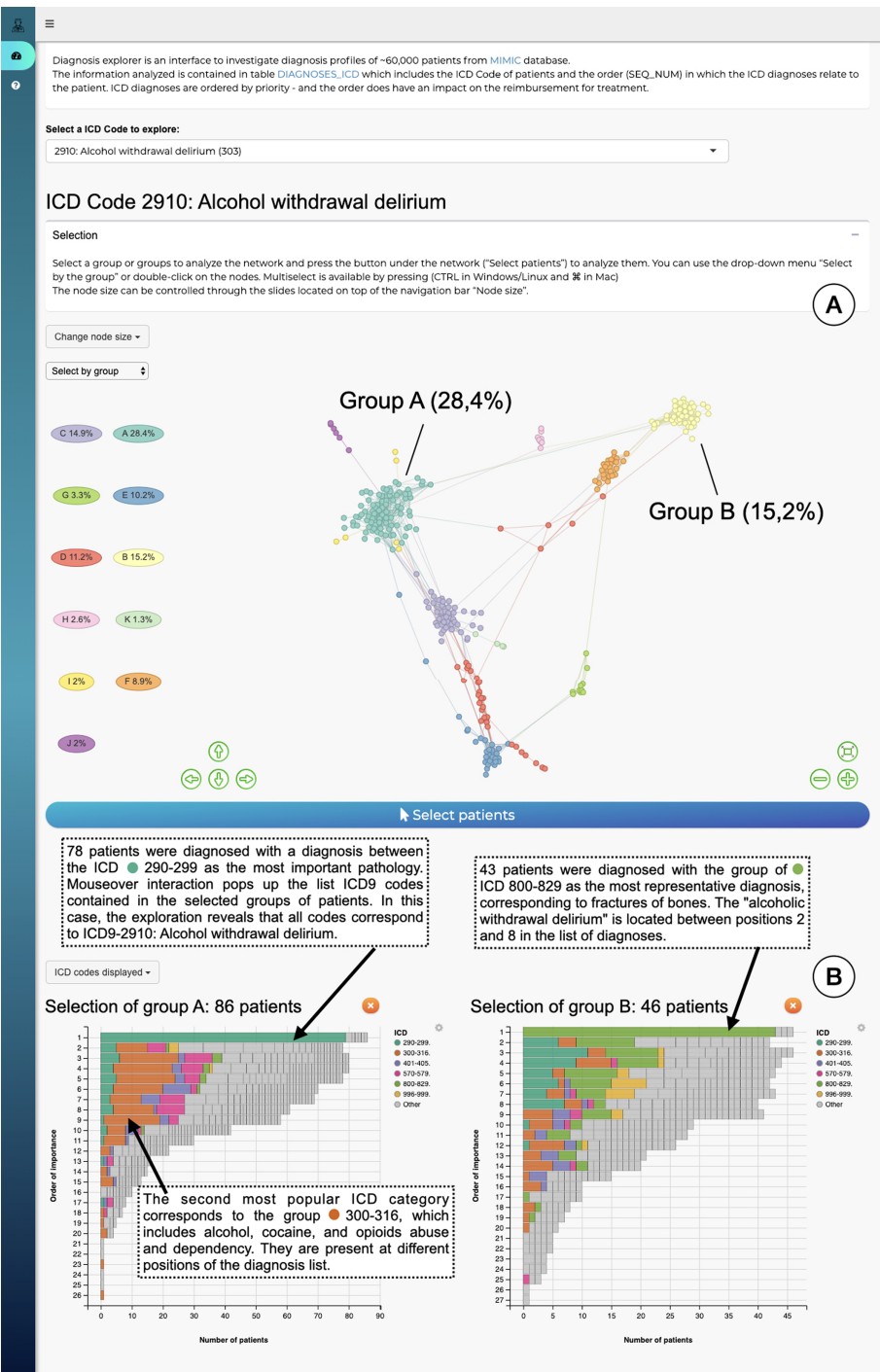

**Figure 4 The interface to explore the diagnosis profiles in the MIMIC-III database.** (A) Network visualization of those patients who have alcohol withdrawal delirium as one of their diagnoses. The network is visualized using a force-directed layout. Node colors are assigned automatically following Louvain community detection. (B) Bar-charts to compare the diagnosis profiles of selected groups in the network. Color corresponds to ICD category. In this example Group A contains patients with alcohol withdrawal delirium as the primary diagnosis; in contrast, Group B lists closed fractures as the most relevant diagnosis, and alcohol withdrawal delirium is only in the 2nd to 8th position.

The Louvain algorithm defines clusters by measuring the density of links inside the group compared to the links between them, which is close to the user interpretation of networks. However, the interpretation of a STAD-R network is not limited to discrete clusters. It aims to represent all relationships between points, including other types of patterns, such as trends or loops. The user can subsequently select either a cluster in this visualisation or individual patients, which will then trigger the display of a barchart which gives more information for that particular cluster (Fig. 4B). This stacked barchart shows how different ICD codes are spread across the different positions in the list of diagnoses: how many patients have code 2910 at the first position in the diagnosis list, how many at the second position, etc.; the same goes for the other ICD codes. Total bar lengths decrease as the position in the list decreases due to the fact that different patients have different lengths of diagnosis lists.

## DISCUSSION

The definition of a custom similarity metric together with a flexible dimensionality reduction technique constitute the key elements of our approach. In this section, we evaluate the benefits of STAD to detect patterns in diagnostic data compared to other popular methods and further discuss the application of the presented distance metric in a different but similar context.

### Comparing STAD to other dimensionality reduction methods

The projection of distances in STAD-R aims to enhance the representation of similarities using networks. Similar groups of patients tend to be inter-connected and perceived as a homogeneous cohort. The outputs of three popular algorithms (MDS, t-SNE, and UMAP) are compared with STAD-R in Fig. 5. The population used in this example is the collection of MIMIC-III patients with alcohol withdrawal delirium (ICD-9 291.0), which was also used for Fig. 4. The MDS projection endeavors to approximate all distances in data within a single 2D plane. Dimensionality methods such as t-SNE and UMAP favor the detection of local structures over the global, although UMAP also retains part of the general relations. Conversely, the abstract graph produced by STAD-R must still be embedded to be visualized, and the selection of the layout may produce slightly different results. Unlike scatterplots, node-link representations provide a more flexible platform for exploring data, especially when node positions can be readjusted according to the analyst and data needs (*Henry, Fekete & McGuffin, 2007*).

In the four plots of Fig. 5, the same points were highlighted to ease the comparison between them. These groups correspond to three communities identified by the Louvain method in the interface. For instance, community 1 and 3 correspond to the patients analyzed in section "Results". Community 1 were patients diagnosed with alcohol withdrawal delirium as the primary diagnosis (Group A in Fig. 4); community 3 are patients with fractures of bones as the primary diagnosis (Group B in Fig. 4); community 2 are patients with intracranial injuries such as concussions. Despite the simple comparison presented, further analysis between these groups confirmed qualitative differences between profiles and a closer similarity between communities 2 and 3 than 1. The initial causes

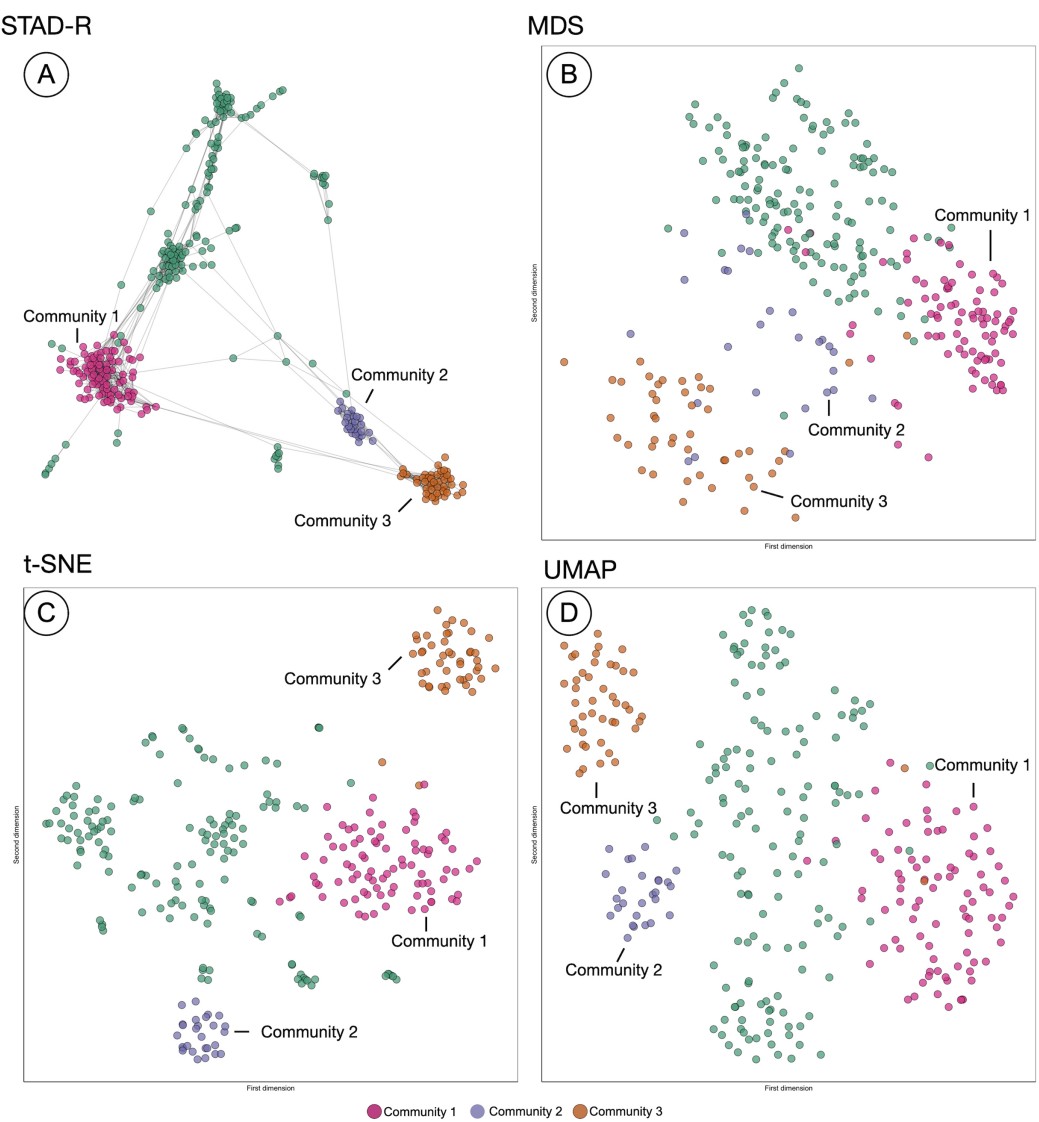

**Figure 5 Comparison of STAD-R, MDS, t-SNE and UMAP using the population of patients with patients with alcohol withdrawal delirium (ICD-9 291.0).** (A) ForceAtlas2 embedding of STAD-R graph; (B–D) MDS, t-SNE and UMAP projections of the same distance matrix used to compute the STAD-R graph, respectively. The three communities were determined by the Louvain algorithm. Community 1 are patients diagnosed with alcohol withdrawal delirium in the first positions of the list. Community 2 were patients with intracranial injuries as concussions. Community 3 are patients with fractures of bones as the primary diagnosis.                         

of communities 2 and 3 are associated with injuries while the primary diagnosis of patients in community 1 is the delirium itself.

In Fig. 5, we can see that communities that are defined in the network (Fig. 5A) are relatively well preserved in t-SNE (Fig. 5C) but less so in MDS (Fig. 5B). However, t-SNE does not take the global structure into account which is apparent from the fact that communities 2 and 3 are very far apart in t-SNE but actually are quite similar (STAD-R and MDS). UMAP (Fig. 5D) improves on the t-SNE output and results in a view similar to MDS.

**Table 2 Objective function in STAD and STAD-R.** The correlation $\rho$ is computed between the original distance matrix $D_X$ and the distance matrix derived from the shortest path graph in $D_U$. The ratio $R$ is calculated from the network at each iteration considering the edges included in the network. Note that distance $d_{\text{network edge}}$ are normalized values between zero and one.

| STAD | STAD-R |
|------|--------|
| $max\ \rho(D_X, D_U)$ | $max\ \rho(D_X, D_U)R = max\ \rho \sum 1 - d_{\text{network edges}} \sum 1 + d_{\text{network edges}}$ |

**Table 3 Distance preservation measures of projections in Fig. 5.** The table describes the Spearman's rank correlation ($\rho_{Sp}$) and the proportion of the first fourteen nearest neighbors preserved (14–nn). The selection of 14 neighbors corresponds to the average cluster size in the MIMIC-III dataset using Louvain community detection. Column "STAD-R graph" represents the abstract graph and column "STAD-R layout" represents the node placement generated by a ForceAtlas2 layout (*Jacomy et al., 2014*) which is the layout implemented in the interface. These results were obtained from a single execution, and stochastic methods such as t-SNE and ForceAtlas2 may provide different values between executions.

| Global/local focus | Measure | MDS | t-SNE | UMAP | STAD-R graph | STAD-R layout |
|--------------------|---------|-----|-------|------|--------------|---------------|
| Global | $\rho_{Sp}$ | 0.54 | 0.41 | 0.47 | 0.52 | 0.47 |
| Local | 14-nn | 0.34 | 0.60 | 0.53 | 0.62 | 0.52 |

Although the interpretation of these visualizations is difficult to assess, quality metrics may help quantify the previous intuitions. Table 2 presents the quantitative measures for global distance and local distance preservation of projections in Fig. 5. Global distance preservation was measured using the Spearman rank correlation ($\rho_{Sp}$). It compares the distances for every pair of points between the original data space and the two-dimensional projection (*Zar, 2005*). Local distance preservations were measured by the proportion of neighbors identified in the projection. This metric quantifies how many of the neighbors in the original space are neighbors in the projection (*Espadoto et al., 2019*). We evaluated this metric using a neighborhood of the first fourteen neighbors, since fourteen is the average cluster size in the MIMIC-III dataset found using Louvain community detection ($14 - nn$).

The richness of the node-link diagram representation of STAD-R cannot be captured using node position in the 2D plane alone. Therefore, STAD-R is analyzed from two perspectives. First, as the abstract graph generated by STAD-R (STAD-R graph) and, second, the two-dimensional projection after graph drawing (STAD-R layout). The abstract graph only considers the connections between nodes to determine the distances between them, whereas the graph drawing results only consider the node placement in the 2D plane.

Based on the values from Table 3, STAD-R obtained equivalent results to other dimensionality reduction methods in the preservation of global and local structures. Although MDS captured global relationships most effectively, STAD-R layout obtained a correlation value equal to UMAP. Local community structure was most effectively captured in the t-SNE layout (at the expense of global structure). Whilst STAD-R's graph is more effective, this local structure is lost on embedding. In comparison with other projection methods, we note that node-link diagrams provide tangible information
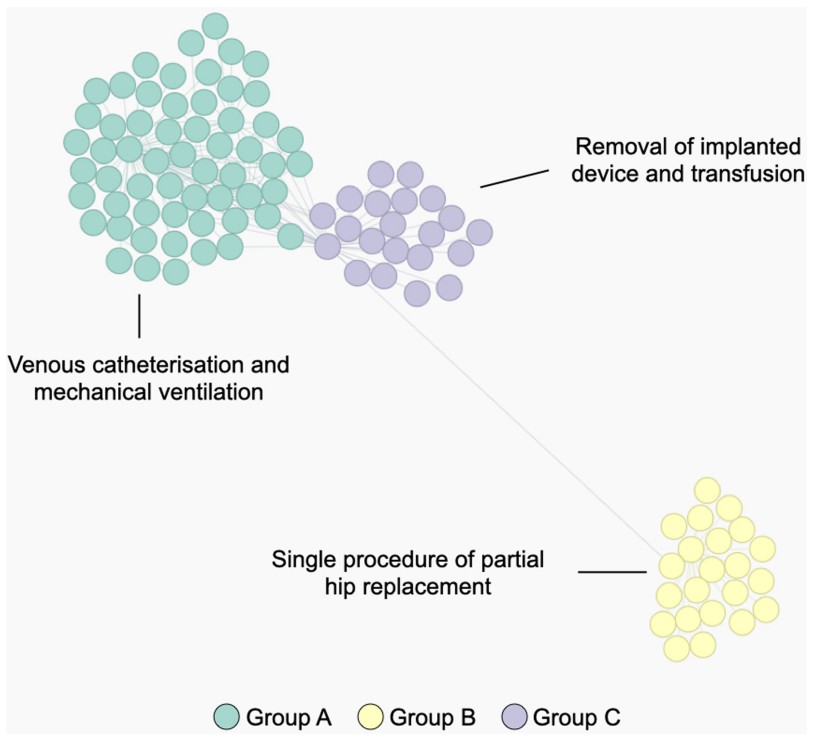

**Figure 6 The population of patients who received a partial hip replacement (ICD 9: 81.52).** The network was computed using STAD-R, and distances were estimated using an adapted version of diagnosis similarity for procedures. Color is based on Louvain community detection.

through links, which enhance the interpretation of relationships and allow thorough exploration through interactions, such as dragging nodes to other positions.

## Similarity measures for ICD procedures

The diagnosis similarity described in section "Diagnosis similarity and distances" is designed for assessing distance between diagnosis profiles, but the principles presented here can be generalized to other terminologies. For example, the procedures which patients receive during a hospital stay are also recorded and also follow an ICD codification: they also contain a list of categories similar to diagnosis. Unlike ICD diagnoses lists, which encodes priority, the order of procedure code lists indicate the sequence in which encode procedures were performed. Thus the weight distribution in the similarity that was used for the diagnosis metric must be adapted to the nature of the procedure data. We can alter the formula to include the relative distance between positions of matched elements instead of the top position in the diagnosis case. Formally, the similarity between two procedure concepts can be then described as follows:

$$M_C(A, B) = \ln\left(1 + \frac{1}{|\text{position}(C_A) - \text{position}(C_B)| + 1}\right)$$

As with diagnosis similarity, the metric is estimated as the sum of individual contributions of matched concepts, $S(X, Y) = \sum_{i=1}^{n} M(X \cap Y)$.

Figure 6 shows a STAD network generated using this adapted similarity for procedures. This example illustrates the population of patients with partial hip replacement (ICD 9: 81.52) in the MIMIC-III population. We can identify three clusters which describe three types of patients: group A are patients with the largest list of activities and are often characterized by venous catheterization and mechanical ventilation; patients in group B are mainly patients with a single procedure of partial hip replacement; patients in group C are characterized by the removal of an implanted device and a blood transfusion (data not shown).

## CONCLUSIONS

In this paper, we introduced a custom distance metric for lists of diagnoses and procedures, as well as an extension to STAD to improve its effectiveness for dissimilar datapoints. The diagnosis similarity measure can be applied to any ordered list of categories in a manner that is not possible with the measures available in the literature so far. The metric is designed to identify differences between patients through standardized concepts (diagnosis and procedures) where the weights of matching concepts are adapted to highlight the most relevant terms. As mentioned in *Boriah, Chandola & Kumar (2008)*, selecting a similarity measure must be based on an understanding of how it handles different data characteristics. The projection of data using STAD-R allows both for the detection of local structures and the representation of the global data structure. While no dimensionality reduction output from a high-dimensional dataset can completely project all relationships in the data, the connection of nodes in the graph allows a granular selection and exploration of cohorts. Furthermore, the embedding of the network into an interactive dashboard provides a level of convenience that supports interpretation of the analysis results of the network.

Moreover, as discussed previously, STAD-R can reveal equivalent data signals at multiple levels to other dimensionality reduction methods. Quantitative and qualitative (user) evaluation of the method can be further extended with other datasets to assess both the information captured by the graph and the benefits of node-links diagrams to represent the similarity between datapoints. In future work, we plan to further explore STAD-R in collaboration with domain experts in diverse case studies. We also plan to build a more robust interface that allows the computation and exploration of STAD-R networks in a tailored environment.

### Funding

This project is financed through the IWT SBO ACCUMULATE Grant nr 150056 and the Flemish Government "Onderzoeksprogramma Artificiele Intelligentie (AI) Vlaanderen" programme. The funders had no role in study design, data collection and analysis, decision to publish, or preparation of the manuscript.

## Grant Disclosures

The following grant information was disclosed by the authors:
IWT SBO ACCUMULATE: 150056.
Flemish Government.

## Competing Interests

The authors declare that they have no competing interests.

## Author Contributions

- Daniel Alcaide conceived and designed the experiments, performed the experiments, analyzed the data, performed the computation work, prepared figures and/or tables, authored or reviewed drafts of the paper, and approved the final draft.
- Jan Aerts conceived and designed the experiments, performed the experiments, analyzed the data, authored or reviewed drafts of the paper, and approved the final draft.

## Data Availability

The code can be found at VDA Lab: http://vda-lab.be/mimic.html.

Interface code can be found at GitHub: https://github.com/vda-lab/ICD_diagnosis_explorer.

The MIMIC-III database is available at MIMIC: https://mimic.physionet.org/.

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
