# Peer review of "A visual analytic approach for the identification of ICU patient subpopulations using ICD diagnostic codes"

_PeerJ Computer Science, doi:10.7717/peerj-cs.430_

## Round 0.1 · original submission · Minor Revisions

Both reviewers agree that your work requires only minor revisions prior to acceptance.

Reviewer 1 raises questions about semantic information loss in the use of rank-based metrics, and the use of patient similarity measures in general. They also ask if a holistic view of the MIMIC database could be achieved with a more complex metric analysed with STAD-R.

Reviewer 2 highlights that figure legends to be made larger, and that the Louvain method employed in this work was not properly introduced. They also highlight text that should be revised for clarity in line 254 (as position in list increases) and line 285 (t-SNE & global structure). Here I also recommend you consider PeerJ's figure instructions (https://peerj.com/about/author-instructions/#figures - specifically, using capitals for multipart figure labels, and not 'highlighting' the subfigure label with a yellow background).

In addition I offer my own comments and suggested revisions:

A. Demonstration application

I found this RShiny app reasonably usable though I found the 'Select Patients' button unintuitive. Also, with some selections it is possible to create an empty plot in the Bar Chart section (this could be due to the minimum length slider setting, but was also after running through several primary ICD codes, subgroups and interactive selecting events, so could be a bug).

Suggestion: It could also be useful to be able to re-select the patient(s) in the network diagram that are shown in a particular chart.

B. Code review

Only the processed data has been provided along with the demo application. Ideally, deployment instructions and scripts for processing the sample data is needed to demonstrate the application can be reused (this also helps since several data items are referred to but not defined in the R code). Inspection of the R code reveals clear comments highlighting different sections, but I note some aspects (e.g. a decision tree feature) are commented out.

1. Please also provide the scripts needed to generate the processed data, or make clear where they can be obtained (eg. in line 241).

Online docs

The equations for the STAD metric appear to not be rendered in mathML, making it somewhat difficult for a reader not familiar with the precise notation used to interpret them.

2. If possible please revise the equation so it renders correctly.


Manuscript.

I have provided an annotated PDF detailing suggested revisions and comments. Questions and critical issues are summarised here:

3. Clarify the use of ICD term weighting in similarity/projection metrics. You note in lines 104-107 that your approach resembles that of Goodall (1966), i.e. that pairs of less common attributes receive higher similarity scores than pairs of common attributes. However, I noted no subsequent evidence that the metric encodes such 'prior-like' behaviour. Please either more clearly explain this aspect, or omit the statement and reference.

4. Line 156 - "what is meant by 'reliable results' - see suggested revisions in PDF.

5. line 164 - the distance metric described probably isn't "new" (i.e. novel), but has perhaps not yet been applied to patent ICD term lists. It may be better to qualify your statement rather than say it is an entirely new metric.

6. The wording and notation used in the equation at line 177 M(c_A,c_B) suggests that c_A and c_B may be *different concepts). Suggest this is revised to M_c(A,B) - since (as I understand it) M is defined for a particular concept (indeed, it is referred to as such in line 179).

7. line 189-192 - these two sentences describing how the bimodal behaviour of the metric combined with the scaling effect of the order leads to a left-skewed distribution should be improved for clarity. Ideally, a reference to this cumulative behaviour could be given, since it is a behaviour common to many similarity measures composed of discrete scores (e.g. pairwise alignments).

8. Line 269-70. Simpler and more formal to simply say 'STAD-R's graph needs to be laid out/embedded to be visualised'.
9. line 270-274. Is the detailed description of alternative 2D layout/embedding methods relevant to the discussion ? I'm also unconvinced that interactive layout is critical for this application without evidence from user evaluation.

10. line 275-293 gives a fairly in-depth description of the appearance of the different louvain clusters under different projection methods as contrasted to the STAD-R graph embedding. The distinctness of the clusters under these projections could be quantified rather than simply described - was this attempted ? A quantitative analysis could also allow effectiveness of each visualisation method to be systematically evaluated across all ICD code graphs.

11. Line 288-293 - it isn't clear to me why the outliers in community G have been described in detail - is this an advantage of the method or a problem with the community analysis technique ?

12. Line 298-300. This could be reworded for clarity. Here I think you mean to explain that unlike ICD diagnostic lists, which encode 'priority', the order of procedure code lists indicate the sequence in which procedures were performed. It's meaningless to say 'the position of a procedure is equally important across the list'.

13. Line 305 - the modified similarity metric for ICD procedures includes | position(C_a)+position(C_b) | - did you mean '-' here ?

14. I was surprised to find there was no discussion concerning planned further work, such as formal user evaluation to quantitate the effectiveness of the approach. I hope that such work will be carried out in the future!

·

Basic reporting

Firstly, the paper was well written. Clear, unambiguous, professional English language used throughout. Secondly, the introduction gave a clear description of the problem and relevant studies. Figures are high quality and well described.

Experimental design

The research question is well defined and meaningful. The experimental was well designed. They also provide an online demo of the subpopulation identification in MIMIC dataset.

Validity of the findings

The real contribution of this paper is the STAD-R. The proposed metric to measure the similarity between diagnosis profiles is not bad for scenario which want to identify subsets in a target population with specific diagnosis. The order or rank of target diagnosis was well utilized to distinguish the subsets. My concern is such a rank-based similarity is not fully taking advantage of the semantic relationship among different diagnosis. In this paper, the author mapped the original ICD code to clinically meaningful sections and it will help to reduce the impact of semantic relationship. The author also discussion about the similarity of procedure based on similar metric. I do not think both of these similarity methods are the perfect way to measure patient similarity. There are some studies consider both the order and the semantic relationship among diagnosis profile. Maybe the author can discuss this.

In this paper, the network generated based on STAD-R is very clear and impressive to distinguish subpopulation with specific diagnosis. I am curious about the results that using a more complicated similarity metric in all MIMIC patients and visualize them using the STAD-R . Hopefully, it will provide a whole map of all patients and can also help to identify subpopulation based on network directly without providing a diagnosis code.

Additional comments

It's a very impressive work. Congratulation to the authors.

Reviewer 2 ·

Basic reporting

In Figure 1 and Figure 3, titles and labels are difficult to read since the font size is small.
In Figure 2, the formulas below networks are difficult to read since the font size is small.
The Louvain method is used to identify patient communities, but it is not explained.

Experimental design

No comment.

Validity of the findings

At row 288, there is a comment about the points near to the first patient community that are not part of the same community as defined by the Louvain algorithm. Is it possible to compare STAD to other dimensionality reduction methods even in terms of number and impact of this kind of disagreement?

Additional comments

At row 254, it is said that “total bar lengths decrease as the position in the list increases”, but in Figure 4 the total bar lengths decrease for lower positions, so towards the bottom.
At row 285, it is not clear if t-SNE does take the global structure into account, or if t-SNE does not take the global structure into account.

---

## Round 0.2 · Minor Revisions

Thank you for submitting your revised manuscript. I was pleased to see several significant additions in response to the reviewers comments on the previous version. However, given the previous decision, I did not consider it necessary to request the previous reviewers to comment on this new version.

In general all comments on the previous version have been addressed in your rebuttal, and through revisions to the manuscript and its accompanying github repository. Still, I note a number of minor revisions to address grammatical issues and suggested rewordings to more clearly communicate your findings. These are described in the attached PDF, along with one significant request:

In lines 304-308 an interpretation of the results of Table 3 is given. This is one of the most substantial of the new additions to the manuscript, and whilst the results are very clear, I found their reporting and interpretation needs work in order for the manuscript to be accepted for publication.

Please refer to the annotated PDF for full details.

---

## Round 0.3 · accepted · Accept

Thank you for considering revisions in the previous round.

Whilst I have recommended 'Accept', please address the following minor points of revision during the proofing stage:

1. Line 109: 'comorbidity diseases' - reword. These are usually referred to as 'comorbidities' or 'disease comorbidities'.

2. Line 173-174 (Table 1). The legend for table 1 seems to actually be the legend for Table 2. Please provide an appropriate legend for Table 1 (and apologies for not asking for this revision in the previous round).

3. Line 267: "Similar groups of patients tend to be inter-connected, which are perceived as a homogeneous cohort." - here I suggest omitting the comma and replacing 'which are' with 'and' to read '..and perceived as a homogeneous cohort'.

Reviewer 2 ·

Basic reporting

No comment.

Experimental design

No comment.

Validity of the findings

No comment.

Additional comments

After this revision, authors have adequately addressed the reviewers’ comments, therefore I think that the paper could be published.